# Glycoalkaloid Composition and Flavonoid Content as Driving Forces of Phytotoxicity in Diploid Potato

**DOI:** 10.3390/ijms24021657

**Published:** 2023-01-14

**Authors:** Katarzyna Szajko, Paulina Smyda-Dajmund, Jarosław Ciekot, Waldemar Marczewski, Dorota Sołtys-Kalina

**Affiliations:** 1Plant Breeding and Acclimatization Institute–National Research Institute, Platanowa 19, 05-831 Młochów, Poland; 2Laboratory of Biomedical Chemistry, Ludwik Hirszfeld Institute of Immunology and Experimental Therapy, Rudolfa Weigla 12, 53-114 Wrocław, Poland

**Keywords:** allelopathy, gene expression, gene ontology, leptine, solamargine, solasonine, transcriptomics

## Abstract

Despite their advantages, biotechnological and omic techniques have not been applied often to characterize phytotoxicity in depth. Here, we show the distribution of phytotoxicity and glycoalkaloid content in a diploid potato population and try to clarify the source of variability of phytotoxicity among plants whose leaf extracts have a high glycoalkaloid content against the test plant species, mustard. Six glycoalkaloids were recognized in the potato leaf extracts: solasonine, solamargine, α-solanine, α-chaconine, leptinine I, and leptine II. The glycoalkaloid profiles of the progeny of the group with high phytotoxicity differed from those of the progeny of the group with low phytotoxicity, which stimulated mustard growth. RNA sequencing analysis revealed that the upregulated *flavonol synthase/flavonone 3-hydroxylase-like* gene was expressed in the progeny of the low phytotoxicity group, stimulating plant growth. We concluded that the metabolic shift among potato progeny may be a source of different physiological responses in mustard. The composition of glycoalkaloids, rather than the total glycoalkaloid content itself, in potato leaf extracts, may be a driving force of phytotoxicity. We suggest that, in addition to glycoalkaloids, other metabolites may shape phytotoxicity, and we assume that these metabolites may be flavonoids.

## 1. Introduction

Allelopathy is an important ecological phenomenon and a type of plant communication with the environment that shapes plant and microorganism communities in both natural ecosystems and agroecosystems. Allelopathy has both positive and negative effects [1]. Influence on the composition and activity of associated bacterial and plant communities reflects the promotion/impediment of soilborne diseases and symbiosis, e.g., migration of beneficial plant-growth-promoting rhizobacteria, regulation of the spread and composition of weeds, and nutrient availability in the soil [2,3].

Allelopathy is a very complex phenomenon and should be considered in a broader context of chemical interactions among organisms involved in soil chemical ecology [1]. In plant–plant allelopathy, the negative interactions are mainly manifested as impaired seed germination or restricted plant growth [4]. However, at low doses, some compounds and allelochemicals can stimulate target plant growth [5]. The main driving force of allelopathy is the release of allelochemicals passively or actively into the soil or as volatiles into the air [6]. Most of these chemicals are secondary metabolites; however, allelopathic nonproteinogenic amino acids such as *m* m-tyrosine and l-canavanine are also products of primary metabolism [7,8,9]. The study of allelopathic interactions under laboratory conditions that are devoid of an edaphic context is considered the study of phytotoxicity (the phytotoxicity of allelochemicals), which is a particular component of allelopathy [1,10].

The genus *Solanum* contains approximately 2000 species grown in many climatic zones and different elevations [11]. Four cultivated potato species and approximately 107 wild relatives are currently recognized [12]. In crop cultivation, allelopathy is an essential component of field interactions, as it is involved in weed management and crop protection [13]. Allelopathy of potato on neighboring plants (weeds) is difficult to observe since the date of potato emergence may pass with the date of weeds’ appearance. Additionally, potato dense canopy can be competitive with weeds [14,15]. However, due to impaired growth of plants in the subsequent season, cultivated potato plants should not be planted after tomato (*Solanum lycopersicum* L.) or pepper (*Capsicum annuum* L.) plants. Such a rotation system is recommended due to the accumulation of allelopathic compounds—glycoalkaloids—in the soil [16]. Thus, potato glycoalkaloids, which are the main cause of soil sickness, may affect soil microorganisms and future crops [16].

Glycoalkaloids (steroidal alkaloids) are bioactive compounds that are produced in *Solanum* spp. and are involved in allelopathic interactions [17,18]. These compounds are composed of aglycon solanidine and carbohydrate moieties. In *Solanum chacoense* leaves, there are two additional classes of glycoalkaloids, namely, leptines and leptinines, composed of the aglycones leptinidine and 23-acetylleptinidine [19]. Glycoalkaloids in aboveground plant parts are involved in host plant resistance, and those that are excreted or leaked from plant organs are involved in plant–plant and plant–microorganism interactions in the soil in potato fields [20]. Our previous analyses also confirmed a substantial role of glycoalkaloids in the phytotoxicity of wild potato relatives, diploid hybrids, and various potato cultivars, with the strongest effects observed among wild species and diploid hybrids [17]. Despite a significant negative correlation between the glycoalkaloid content in potato leaf extracts and phytotoxicity, some of the genotypes with high glycoalkaloid contents have low phytotoxicity [17]. The paradox of why high glycoalkaloid contents do not always reflect high phytotoxicity remains unexplored.

Currently, research on phytotoxicity focuses on the identification of target processes that are affected by allelochemicals in acceptor plants or the isolation of specific compounds from donor plants. Despite their advantages, biotechnological and omic techniques have not been applied to characterize phytotoxicity in depth. Although genetic studies on allelopathy/phytotoxicity have been carried out [21,22,23], there are only a few studies that recognize the genetic basis and global gene expression changes involved in these processes [18,24,25]. To our knowledge, no studies have been conducted to determine the genetic or molecular origins of positive allelopathic interactions, despite the critical role that these origins may play in the ability of allelopathic interactions to shape the yield of crops and their competitiveness against weeds.

Here, we show the distribution of phytotoxicity and glycoalkaloid content in a segregating population obtained from crossing diploid potato hybrids with *S. chacoense* and try to clarify the source of phytotoxicity variability among plants whose leaf extracts have high glycoalkaloid contents. We also recognize the transcriptomic and metabolomic aspects of potato phytotoxicity that may affect the response of the test plant mustard (*Sinapis alba* L.) to potato leaf extracts. We focus particularly on individuals who, despite having high glycoalkaloid content in their leaf extracts, which seemed to be the driving force of potato phytotoxicity, were weakly or highly phytotoxic or even stimulated the growth of the test plant (exhibited positive interactions). Finally, we provide new insight into the possible source of phytotoxicity variation in potato with high glycoalkaloid contents against mustard as the acceptor plant species.

## 2. Results

### 2.1. Evaluation of the Glycoalkaloid Content and the Phytotoxicity of Leaf Extracts from Individuals of the Potato Population

Potato phytotoxicity was evaluated based on aqueous leaf extracts. Water is the preferred solvent for the extraction of allelochemicals that are intended for biological analyses, since these extracts most closely resemble the leaching of allelochemicals from plants by rain or dew [1]. The glycoalkaloid content and phytotoxicity were evaluated for 113 individuals of the potato population (Table 1). The glycoalkaloid content ranged from 1.1 to 46.7 μg mL^−1^ (Table 1), and the phytotoxicity varied from 41% to 140%, where values greater than 100% indicated growth stimulation (Table 1, Figure 1). Neither the glycoalkaloid content nor the phytotoxicity were normally distributed (Appendix A), and no significant correlation was found between the glycoalkaloid content and the phytotoxicity (r = −0.155, *p* = 0.0568). All 113 individuals were assigned to groups A-F based on the criteria described in the M & M (Table 2). The highest percentage (44%) of the individuals were assigned to group C, where low glycoalkaloid contents and high phytotoxicity were observed in those members (Table 2). The lowest percentage corresponded to individuals of group A (4%), whose members had high glycoalkaloid content and high phytotoxicity.

### 2.2. Profiles of Differentially Expressed Genes (DEGs)

To identify which genes may influence the phytotoxicity of potato, bulk samples were constructed as described in the M&M, which were named A’, B’, and F’. We performed comparisons of bulk samples A’ and B’ and of A’ and F’. The total number of raw reads in bulk samples A’, B’, and F’ ranged from 28,683,902 to 28,928,669. There were 6044 and 6320 statistically significant DEGs in the A’ vs. B’ and A’ vs. F’ comparison groups, respectively. After a sharp cutoff (false discovery rate (FDR)-adjusted *p* value < 0.05 and 2.0 ≤ log_2_ (fold-change) (Log_2_FC ≤ −2.0), 866 DEGs remained in the A’ vs. B’ library comparison group, and 692 were left in the A’ vs. F’ library comparison group (Appendix A). The ten most abundant transcripts of upregulated and downregulated genes are listed in Table 3. Among the upregulated genes, in the A’ vs. B’ comparison group, *flavonol synthase/flavanone 3-hydroxylase-like* (*FLS*) was the most highly expressed gene (Log_2_FC = 8.30), and *chitin-binding lectin 1-like* was the most downregulated (Log_2_FC = −10.00) (Table 3). In the A’ vs. F’ comparison group, *probable trans-2-enoyl-CoA reductase mitochondrial* was the most upregulated gene (Log_2_FC = 7.07), while *cytochrome P450 72A15-like* was the most downregulated one (Log_2_FC = −9.85) (Table 3). *FLS* was also recognized as an upregulated gene in the comparison of bulk samples A’ and F’ (Log_2_FC = 6.40).

Among 1558 significant DEGs detected after a sharp cutoff in the A’ vs. B’ and A’ vs. F’ comparison groups, only 209 DEGS were found to be common in both groups (Figure 2).

### 2.3. Gene Ontology (GO) Analysis of Differentially Expressed Genes (DEGs)

GO analysis was performed to classify the DEGs derived from the A’ vs. B’ and A’ vs. F’ comparison groups and divided these DEGs based on their biological process (BP), cellular component (CC), and molecular function (MF). The top ten most abundant GO terms are shown in Figure 3A,B. All the significant GO terms are listed in Appendix A. The most significant GO terms for the DEGs in the A’ vs. B’ comparison group in the BP category were *photosynthesis, oxidation–reduction process*, and *secondary metabolic process*; the CC category, *photosystem*, *thylakoid membrane*, and *photosynthetic membrane*; and the MF category, *oxidoreductase activity*, *chlorophyll binding*, and *tetrapyrrole binding* (Figure 3A). The most significant GO terms for the DEGs in the A’ vs. F’ comparison group in the BP category were *cell wall organization or biogenesis*, *carbohydrate metabolic process*, and *photosynthesis*; the CC category, *extracellular region*, *photosystem*, and *external encapsulating structure*; and the MF category, *tetrapyrrole binding*, *oxidoreductase activity*, and *chlorophyll binding* (Figure 3B). The subgraph in BP induced by the ten most significant GO terms is presented in Appendix A.

### 2.4. Flavonol Synthase/Flavanone 3-Hydroxylase-like (FLS) Expression and Total Flavonoid Content Analyzes

We checked whether *FLS* expression levels differed in the individuals assigned to bulk samples A’, B’, and F’ (Figure 4, grey bars). Ranging from 5.8 × 10^−5^ to 7.7 × 10^−4^, the transcript levels of *FLS* in the individuals of A’ were the lowest, and these levels were significantly different from those in individuals assigned to the B’ and F’ samples. Among the B’ individuals, the transcript levels were between 2.2 × 10^−2^ and 7.3 × 10^−2^ and between 1.9 × 10^−2^ and 3.3 × 10^−2^ in the plants composing F’ (Figure 4, grey bars). The *FLS* upregulation, which was observed in bulk samples, was repeated in individuals characterized as having low phytotoxicity or stimulating mustard growth (Figure 4, grey bars). The total flavonoid content in the individuals of the A’ sample was significantly lower than in the B’ and F’ samples (Figure 4, blue dots). No significant differences were observed in total flavonoid content between samples B’ and F’.

### 2.5. Glycoalkaloid Composition in Bulk Samples A’, B’, and F’

To recognize glycoalkaloids that are represented in bulk samples A’, B’, and F’, we performed LC–MS analyses. Six glycoalkaloids were recognized (Table 4). Four glycoalkaloids were derived from the aglycon solanidine: solasonine, solamargine, α-solanine, and α-chaconine; one glycoalkaloid was derived from the aglycone leptinidine: leptinine I; and one glycoalkaloid was derived from 23-acetylleptinidine: leptine II (Table 4). In bulk sample A’, all six glycoalkaloids were present. α-Chaconine was the most abundant glycoalkaloid in bulk samples A’, B’, and F’ (52.8%, 57.7%, and 57.9%, respectively). Bulk samples B’ and F’ had similar glycoalkaloid compositions and frequencies, where α-solanine, α-chaconine, and leptinine I were present.

## 3. Discussion

A comprehensive meta-analysis of allelopathy confirmed that the effects of allelopathy on the germination and growth of test plants are more negative under controlled laboratory conditions (phytotoxicity) than under conditions that resemble natural ones [26]. It is highly possible that potato phytotoxicity induces responses in target plants that differ from those that may be observed during allelopathic interactions in the field; however, it cannot be inferred that allelopathy in potato fields is an inexistent phenomenon. Glycoalkaloids leached from potato tubers or from plant residues can accumulate in the soil (up to 0.6 kg ha^−1^) and remain there in high amounts until spring, thus affecting the following crops and soil microorganisms [27]. In the present study, we provide new insights into the quantitative nature of the phytotoxicity of potato plants.

In this study, we used the potato diploid population 15–1 to explore the phytotoxic effects of aqueous leaf extracts of potato on mustard. In the population, the phytotoxicity trait segregates and is highly variable. Extreme phenotypes that inhibit mustard growth by up to 59% as well as those that stimulate growth by up to 40% are recognized. The distribution of phytotoxicity deviates from normality and resembles a bimodal distribution, where individuals with moderate to high phytotoxicity and those that stimulate mustard growth constitute the majority. This is in contrast to the findings of other studies, where the distribution of phytotoxicity was close to normal and individuals with moderate effects constituted the majority [21]. Additionally, only negative effects on the growth of the test plants were observed [21]. Allelopathy/phytotoxicity is a complex trait that depends on many factors related to donor and acceptor plants; thus, owing to complex polygenic inheritance, this is an example of a quantitative trait, which is the outcome of relatively small effects of multiple genes [22,23,28]. The quantitative nature of allelopathic interactions has previously been recognized as a competitive advantage over neighboring plant species. It is estimated that variation in allelopathy within closely related species or individuals of the same species is significantly lower than that between distinct species [26]. However, intraspecific variation in allelopathy is common and is mainly related to plant habitat, as is the case for sumac (*Rhus tripartita*) growing in various locations [29]. Variation in allelopathy may also be associated with genotype, e.g., variation could be related to a specific mutation, as was shown in *Arabidopsis thaliana* accessions in which the absence of the biosynthetic pathways of an indolic glucosinolate hydrolysis product was found to cause the variation [30]. In all these cases, regardless of the source of variability, allelopathy is based on different compositions of metabolites, which ultimately constituted a specific fingerprint and characterized the plant phenotype [29,30]. Therefore, we argue that in the case of the 15–1 population, the variability of phytotoxicity is mainly genotype-dependent.

Since glycoalkaloids are the main secondary metabolites in potato and are direct causes of soil fatigue after potato cultivation, we analyzed the glycoalkaloid content in leaf extracts of all the individuals in the potato population. Interestingly, the correlation between the glycoalkaloid content and the phytotoxicity was not statistically significant. This result is in contrast with that of our previous study [17], in which glycoalkaloid content was correlated with phytotoxicity in a group of potato genotypes that had no common origin and constituted various ploidy levels (cultivars, diploid hybrids, and wild species). In the progeny of a half-sib family, these relations may be different given a more homogeneous genetic background. *S. chacoense* was used in this study as a male parent. In *S. chacoense*, the glycoalkaloid content segregates in individuals among progeny generated by crossing [31]. Based on phytotoxicity and glycoalkaloid content, individuals were assigned to six groups to observe the relationships between these two traits in the population (Table 2). A group of plants with low glycoalkaloid content and high phytotoxicity constituted 44% of the progeny, which could be the reason for the nonsignificant correlation in this population.

We hypothesized that various glycoalkaloid compositions among the progeny may be reflected in the different mustard responses to the direct phytotoxic effects of potato leaf extracts.

To more effectively identify genes and glycoalkaloids that may underlie phytotoxicity, we analyzed bulk samples consisting of individuals with high glycoalkaloid content that affect mustard in three different ways: induce high phytotoxicity (bulk sample A’), induce very low phytotoxicity (bulk sample B’), and stimulate growth (bulk sample F’). Recently, there has been little interest in exploring the genetic and molecular basis of positive effects between plants despite numerous studies on a general description of positive effects, such as those in agricultural intercropping systems [32,33]. Here, we also explore positive effects.

The gene expression patterns differ when samples of high phytotoxicity are compared with those of low phytotoxicity or those that stimulate mustard growth. Indeed, only 13.5% of DEGs were common among these different comparison groups (Figure 2). Among the ten most abundant transcripts between these comparisons, the upregulated gene *FLS* was common (Table 3). *FLS* is a key enzyme of the flavonoid biosynthetic pathway. Several *FLS* homeologs encode isoforms displaying bifunctional activity, *FLS* activity and flavanone 3-hydroxylase (F3H) activity, all of which are relevant for flavonol accumulation in plant tissues [34].

GO analysis classified the genes that were differentially expressed between the bulk comparison groups: high phytotoxicity vs. low phytotoxicity (A’ vs. B’) and high phytotoxicity vs. stimulating mustard growth (A’ vs. F’). We induced subgraphs for the ten most significant GO terms. In the BP category, although most of the GO terms were classified under the higher-level node, namely, *cellular process* and *metabolic process*, both comparisons showed differences. In the common node *metabolic process* and for terms that are grouped under the node *photosynthesis,* however, many GO terms in A’ vs. B’ refer to *secondary metabolic process*, while in A’ vs. B’, they refer to *carbohydrate metabolic process*. There is also a third, additional node that differentiates both comparisons: *response to stimulus* (in A’ vs. B’) and *cellular component organization or biogenesis* (in A’ vs. F’). This indicated a metabolic shift between the potato groups, and such differences in metabolism could be a source of different physiological responses in mustard. This may also explain changes in the composition of secondary metabolites, such as glycoalkaloids. All the analyzed bulk samples have high glycoalkaloid contents; nevertheless, the samples with high phytotoxicity (A’) significantly vary in glycoalkaloid composition and frequency from samples with low phytotoxicity (B’), which stimulates growth (F’). Sample A’ consists of the six recognized glycoalkaloids (α-solanine, α-chaconine, leptinine I, leptine II, solasonine, and solamargine) and have high phytotoxicity. In samples B’ and F’, leptine II, solasonine, and solamargine are absent (Table 4). It was previously noted that solasonine and solamargine present in a mixture may act synergistically and inhibit plant growth more strongly than each compound separately could [35]. When solanine and solamargine are present simultaneously in their leaf extracts, individuals are highly phytotoxic despite having a low glycoalkaloid content [18]. Thus, the lack of solasonine and solamargine in B’ and F’, rather than the various percentages of each glycoalkaloid, is involved in the low phytotoxicity of these samples. We concluded that the glycoalkaloid composition, not the glycoalkaloid content in general, may play a leading role in potato phytotoxicity. Glycoalkaloids biosynthesis is governed by the GAME gene family and at least 38 additional genes that are co-expressed with GAME genes, including genes of the mevalonic acid or sterol biosynthesis pathways [31]. Here, transcriptomic data indicate that in A’, B’, and F’ samples, various types of glycosyltransferases may affect glycoalkaloid composition. Glycosyltransferases transfer the sugar moiety from one compound to another. In the case of glycoalkaloids, e.g., the transfer of rhamnose to α-solanine produces α-chaconine. In the comparison A’ vs. B’, putative *UDP-rhamnose:rhamnosyltransferase 1* (Log_2_FC = 4.80) and in the comparison A’ vs. F’, *rhamnose:beta-solanine/beta-chaconine rhamnosyltransferase* (Log_2_FC = −3.03) are most pronounced glycosyltransferases. Thus, different compositions of glycoalkaloids within the groups could be caused by both, various activities of glycosyltransferases and specific allele configurations of the genes involved in glycoalkaloids biosynthesis. Further analyses should include the recognition of glycoalkaloids in all individualsof a population and correlations between specific glycoalkaloids and phytotoxicity.

Individuals with low phytotoxicity and those that stimulate mustard growth have similar compositions and abundance of glycoalkaloids despite inducing various responses in the target plants. The data suggest that, apart from glycoalkaloids, other metabolites may drive phytotoxicity; we suppose that these other metabolites could in particular be flavonoids since a higher level of *FLS* expression found in these individuals was associated with higher levels of total flavonoid content. Specific mutations in flavonoid-related genes give rise to various metabolic profiles of flavonoids that differ in their content and composition [36]. Quantitative and qualitative changes in flavonoids may reflect plant physiological processes and responsiveness to abiotic stresses and biotic interactions, including allelopathy [37,38]. We previously noted that the addition of the flavonoid quercetin to a solution of α-solanine significantly reduced its negative effect on mustard growth [17]. In this case, due to their radical-scavenging abilities, flavonoids can act as antioxidants [39], counteracting the negative effects induced by glycoalkaloids in target plants. Despite groups B and F possessing similar glycoalkaloid composition and similar flavonoid content, the flavonoid composition or quantity of a particular compound of the highest biological activity may be different. Thus, we suppose the major difference between groups B and F that affects phytotoxicity is the different composition of flavonoids and/or other compounds that are present in the potato extract.

Our results confirmed that phytotoxicity in the studied segregating potato population is complex and related to both genetic and metabolic factors. However, due to the quantitative nature of the trait, identifying a single gene or process that would be fully responsible for the phenomenon with such a complex trait as phytotoxicity is practically impossible. The differences in phytotoxicity among individuals of the population most likely arise from the different expression patterns of genes that are involved in the biosynthesis of secondary metabolites. Such differences may be translated into various metabolite profiles in each individual plant. Prediction of phytotoxicity in a population after the crossing of parents with extremely contrasting phenotypes is difficult since high glycoalkaloid contents are not always associated with high phytotoxicity. Thus, we concluded that the composition of glycoalkaloids in potato leaf extracts may be a driving force of phytotoxicity rather than the total glycoalkaloid content itself. Screening of populations obtained in breeding programs could be extended to agroecological traits such as phytotoxicity. Our study revealed the possible involvement of *FLS* in this phenomenon, which has not been described in recent allelopathy/phytotoxicity studies in plants. It is also highly desirable to extend this study to metabolite profiling, in particular flavonoid profiling, in all individualsof the population to identify a direct link between genotype and metabolic phenotype. This could serve as a starting point that will enable the development of management strategies in an agroecological context for the sustainable selection of germplasm to improve crop yields and competitive advantages against weeds.

## 4. Materials and Methods

### 4.1. Plant Material

The potato diploid population 15–1 (progeny, *n* =  113) was derived from a cross of the diploid *Solanum* hybrid DG 88–89 (female parent, generated at the Plant Breeding and Acclimatization Institute, National Research Institute, Młochów, Polandand the wild species *S. chacoense* (male parent, accession POL003:333133 obtained from National Centre for Plant Genetic Resources, Radzików, Poland). DG 88–89 has a low glycoalkaloid content (5.2 μg mL^−1^) and low phytotoxicity (99%), while *S. chacoense* has a high glycoalkaloid content (55.6 μg mL^−1^) and high phytotoxicity (30%). The potato diploid population in 2016 was grown in a greenhouse in triplicate, and at full anthesis, 5 g of fully expanded leaflets from the middle of the plant was collected from each of 113 individuals, immediately frozen in liquid nitrogen, and stored at −80 °C until use.

### 4.2. Evaluation of The Phytotoxicity of Potato Leaf Extracts

One gram of ground potato leaf tissue from three clones of each individual of the 15–1 population was added to 100 mL of distilled water (1% *w*/*v* leaf extract) and shaken for 12 h on a laboratory shaker. The plant tissue was separated from the extract using filter paper. The phytotoxicity was evaluated against the test plant, mustard cv. Rota in three repetitions of each of the three extracts of the individual (*n* = 9). Mustard was chosen for this test since it is tolerant to allelochemicals cultivation [40,41] and is used as a cover crop in potato. Germinated seeds of mustard were transferred into Petri dishes filled with filter paper soaked in water (control) or in potato leaf extracts from each individual of the 15–1 population and incubated for 5 days. The lengths of the control seedlings and those incubated with potato leaf extracts were measured. Phytotoxicity is described as the percentage of the length of the seedlings treated with potato leaf extract in relation to the length of the water-treated control plants, where a percentage greater than 100% indicates growth stimulation and a percentage less than 100% indicates growth inhibition.

The glycoalkaloid content in the potato leaf extracts of each individual of 15–1 population was determined using the colorimetric method according to [42], with modifications as described by [17].

Based on the phytotoxicity of and glycoalkaloid content in the potato leaf extracts of all 113 individuals of the 15–1 population, individuals were assigned to groups A-F according to the following criteria: high, glycoalkaloid content ≥20 µg mL^−1^ and phytotoxicity ≤80%; low, glycoalkaloid content <20 µg mL^−1^ and 100%≥ phytotoxicity >80%; and growth stimulation ≥101%. Groups A, B, and F were selected for further analysis. Plants in these groups were characterized by high glycoalkaloid content in their leaf extracts and had various levels of phytotoxicity: members of group A had high phytotoxicity, members of group B had low phytotoxicity, and members of group F stimulated mustard growth.

### 4.3. Construction of Bulk Samples

Three bulk samples were constructed: A’, B’, and F’. Each bulk sample was prepared from the leaves of three individuals from groups A, B, and F. Such an approach allows for a transition from a population-level scale to selected individuals representing particular phenotypes in the population [43].

The leaf samples were ground in liquid nitrogen, mixed together to form bulk samples. All bulk samples A’, B’, and F’ were used for the analysis of the glycoalkaloid content in the potato leaf extracts. The samples were subsequently subjected to RNA sequencing (RNA-seq), *FLS* expression, and glycoalkaloid composition analyses.

### 4.4. RNA-Seq and GO Analyses of Bulk Samples

For RNA-seq analyses, bulk samples A’, B’, and F’ prepared as described in Section 4.3 were used. RNA from the bulk samples was isolated using TRIzol reagent according to the methods of [44]. cDNA libraries were subsequently prepared using a Dynabeads^®^ mRNA Purification Kit (Ambion, 61,006) and a MGIEasy RNA Directional Library Prep Set (MGI, 1000006386), both of which were used according to the manufacturers’ protocols. The established cDNA libraries were sequenced on a BGISEQ−500 sequencing platform (BGI Genomics, China) by staff at Genomed^®^ (Warsaw, Poland) and subjected to bioinformatic treatment as described by [18]. The DEGs were identified after comparing A’ and B’ and A’ and F’. The results were expressed as Log_2_FC, and DEGs were with an FDR-adjusted *p*-value of <0.05 and 2.0 ≤ log_2_FC ≤ −2.0 were considered significant.

GO analysis was performed using Trinotate software, which followed the translation of the gene nucleotide sequences into amino acid sequences (Transdecoder software) and their annotation in the Swiss-Prot database. Using the topGO package, we assigned the GO terms with a significantly increased frequency of occurrence among DEGs in the A’ vs. B’ and A’ vs. F’ comparison groups according to the *classic* method.

### 4.5. Expression Level of FLS and Total Flavonoid Content

Leaves of three individuals assigned to bulk samples A’, B’, and F’ were ground individually in liquid nitrogen. Total RNA was isolated using a Total RNA Mini kit (A&A Biotechnology, Poland 031–100) according to the manufacturer’s protocol. cDNA was synthesized using a Maxima First Strand cDNA Synthesis Kit for RT–qPCR (Thermo Fisher Scientific, K1672) equipped with dsDNase. The expression of the *FLS* gene (LOC102585933) was measured using RT–qPCR. The primer sequences used for the target gene were 5′-TATCCCTGGCACTTTTGTTGTC-3′ (forward) and 5′-TTGGGCTTTAATATAGTCCTTGTA-3′ (reverse). Potato β-tubulin was used as a reference gene, and the primer sequences were described by [45]. SYBR Green PCR Master Mix (Roche, Switzerland) and 96-well plates in conjunction with a LightCycler 480 II system (Roche, Switzerland) were used. One microliter of cDNA corresponding to 50 ng of total RNA was taken for analysis of each sample. The thermal cycling conditions were as follows: 15 min of denaturation at 95 °C, followed by 40 cycles of 10 s at 95 °C, and 30 s at 60 °C. To confirm the amplification of gene-specific products, the PCR product melting point was determined in the range of 68–95 °C. Three technical replicates of the progeny were included. Relative expression levels were calculated in Microsoft Excel 2010. *T*-tests for ΔΔcycle threshold (Ct) values and calculations of standard errors (SEs) of the means were performed and determined with Statistica software (StatSoft, Inc., Poland) Tulsa, OK, USA).

Total flavonoid content was measured in potato leaf extracts of individuals assigned to bulk samples A’, B’, and F’ according to the method of [46] in modification as described in [17]. Briefly, to 0.5 mL of 1% potato extracts, the following reagents were added sequentially: 1.5 mL of 96% ethanol, 0.1 mL of 10% aluminum chloride (*w*/*v*), 0.1 mL of 1 M potassium acetate, and 2.8 mL of distilled water. Absorbance was measured after 40 min of incubation at room temperature on the Hitachi U−1900 (Tokyo, Japan) spectrophotometer at a wavelength of 415 nm. The total flavonoids were expressed as an equivalent of quercetin.

### 4.6. Analysis of The Glycoalkaloid Composition in Potato Leaf Extracts

Water extracts (1% *w*/*v*) were prepared from the A’, B’, and F’ bulk samples as described in Section 4.2 to determine the glycoalkaloid composition. To calculate the percentage of glycoalkaloids recovered, α-solamarine was added as an internal standard to each potato leaf extract sample (final concentration of 10 ng μL^−1^). The control sample consisted of α-solamarine at the same concentration as that dissolved in water. A total of 750 μL of acetonitrile acidified with 1% formic acid was added to the same amount of potato leaf extract sample and passed through a sterilizing filter (0.2 μm, Nalgene™). Then, the glycoalkaloid fraction was isolated using the solid phase associated with the QuEChERS (UTC) technique. The supernatant was diluted 10-fold with methanol. HPLC–MS analysis was performed on a Dionex 3000 RS-HPLC equipped with a DGP−3600 pump, a WPS-3000 TLS TRS autosampler, a TCC-3000 RS column compartment (Dionex Corporation, CA, USA), and a Bruker micrOTOF-QII mass spectrometer (Bruker Daltonics, Bremen, Germany). The chromatographic column was a 50 × 3.1 (i.d.)-millimeter Thermo Scientific Hypersil GOLDc column with 1.9 μm particles (Part No. 25002–052130, Serial No. 0110796A6, Lot No. 10922).

The results are expressed as the frequency of each compound in the glycoalkaloid fraction found in the sample.

## Figures and Tables

**Figure 1 ijms-24-01657-f001:**
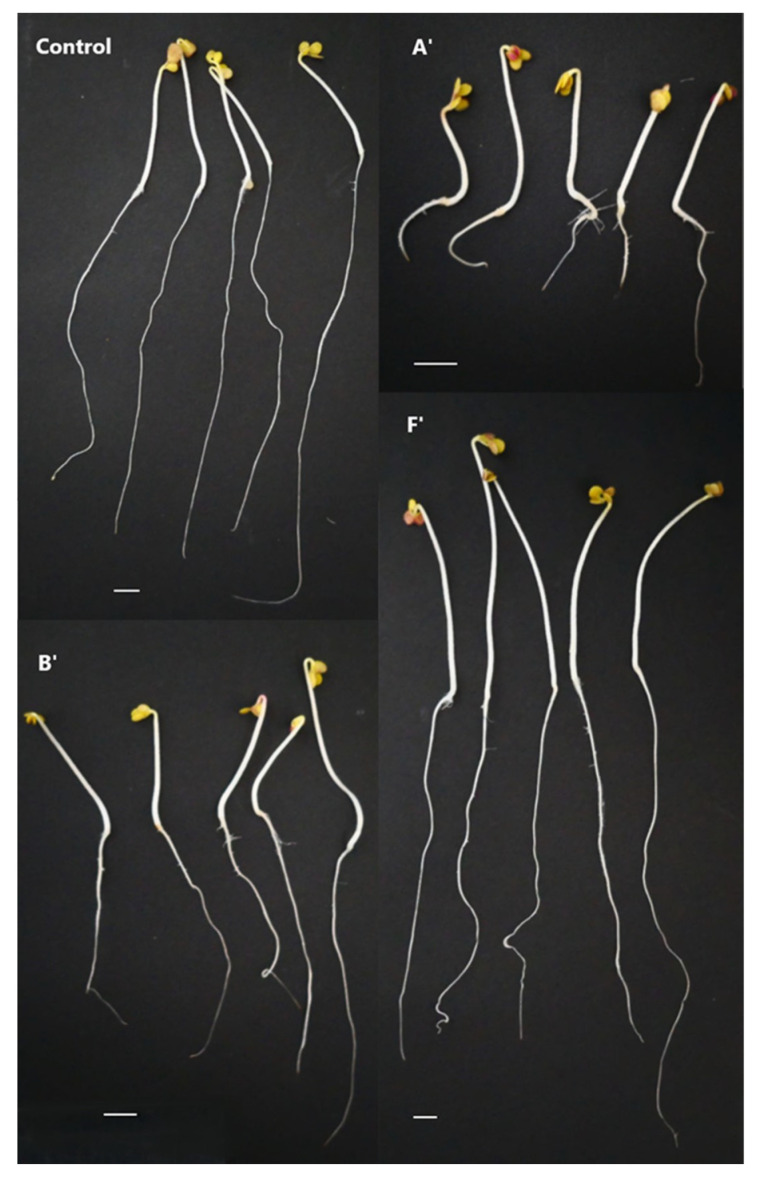
Mustard seedlings that grew for 5 days in water (control) and in potato leaf extracts of A’, B’, and F’ bulk samples. Bars: 1 cm.

**Figure 2 ijms-24-01657-f002:**
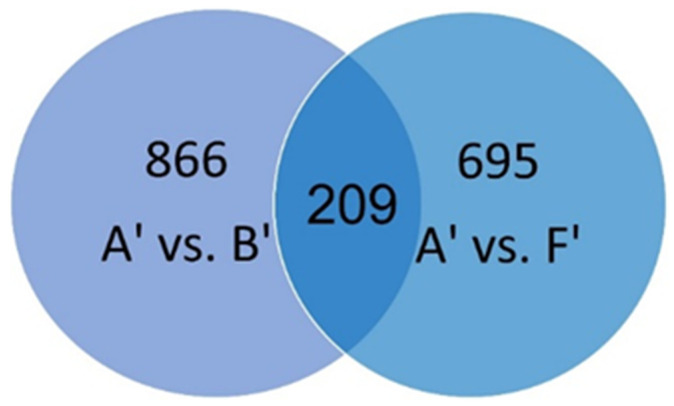
Venn diagram showing the relationships between DEG groups after comparisons of A’ vs. B’ and A’ vs. F’ after a sharp cutoff (FDR-adjusted *p* value < 0.05 and 2.0 ≤ Log_2_FC ≤ −2.0).

**Figure 3 ijms-24-01657-f003:**
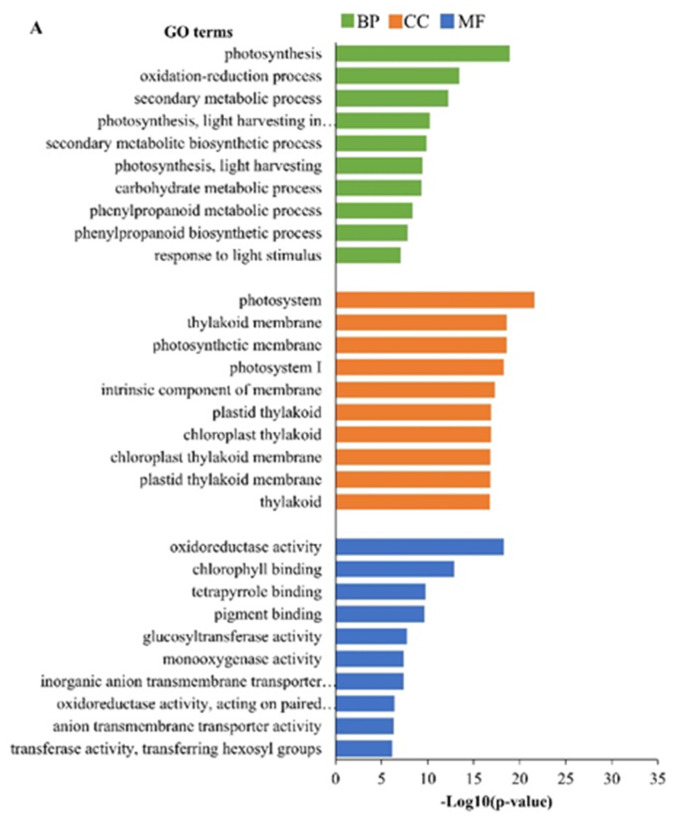
Top ten most enriched GO terms (*p* value ≤  0.05) of DEGs in the (**A**) A’ vs. B’ and (**B**) A’ vs. F’ comparison groups. The green, orange, and blue colors represent the GO terms belonging to biological processes (BP), cellular components (CC), and molecular functions (MF), respectively.

**Figure 4 ijms-24-01657-f004:**
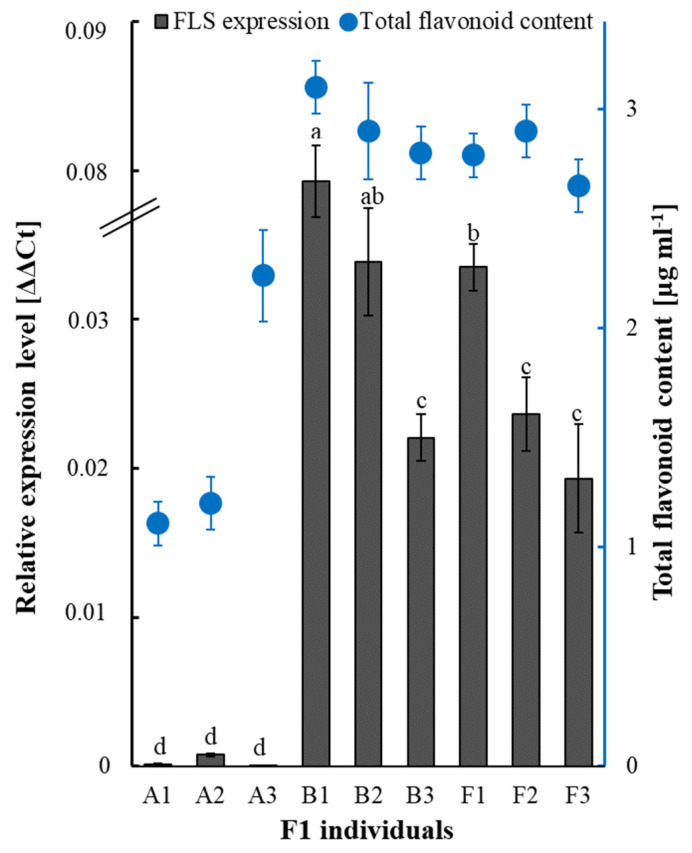
Relative expression level of the *FLS* gene (left axis) and total flavonoid content (right axis) in F1 individuals assigned to bulks A’, B’, and F’. a, b, c, d: homogeneous groups for *FLS* expression level (post hoc Tukey’s honestly significant difference (HSD) test); bars: means ± SD.

**Table 1 ijms-24-01657-t001:** Summary of the glycoalkaloid content and phytotoxicity distribution in the 113 individuals.

Trait	Average	Minimum	Maximum	±SD	Trait Distribution ^1^	
					W Value	*p*
Glycoalkaloid content [μg mL^−1^]	10.764	1.063	46.653	10.943	0.806	<0.000
Phytotoxicity * [%]	86.743	41.618	140.227	28.283	0.934	<0.000

^1^ Shapiro–Wilks test, * Expressed as % of control seedling length.

**Table 2 ijms-24-01657-t002:** Characteristics of groups A-F recognized in population 15–1 based on the glycoalkaloid content and phytotoxicity.

Group	Glycoalkaloid Content ^1^	Phytotoxicity	Number of F1 Individuals (Frequency)
A	High	High	5 (4%)
B	High	Low	8 (7%)
C	Low	High	50 (44%)
D	Low	Low	10 (9%)
E	Low	Growth stimulation	30 (27%)
F	High	Growth stimulation	10 (9%)

^1^ high = glycoalkaloids ≥ 20 µg mL^−1^ and phytotoxicity ≤ 80%; low = glycoalkaloids < 20 µg mL^−1^ and 100% ≥ phytotoxicity > 80%; and growth stimulation ≥ 101%.

**Table 3 ijms-24-01657-t003:** The ten most abundant transcripts of upregulated and downregulated genes according to the RNA-seq data following a comparison of bulk samples A’ vs. B’ and A’ vs. F’.

Locus	Log_2_FC	*p*-Value ^2^	Product Name
A’^1^ vs. B’			
** *upregulated DEGs* **			
LOC102585933	8.30	9.63 × 10^−36^	**flavonol synthase/flavanone 3-hydroxylase-like**
LOC102594045	7.96	1.27 × 10^−10^	UDP-glycosyltransferase 71E1-like
LOC102584518	7.05	2.60 × 10^−25^	metal transporter Nramp5-like
LOC107057710	6.93	1.12 × 10^−7^	uncharacterized protein At1g18380-like
LOC102579990	6.87	1.04 × 10^−7^	uncharacterized protein LOC102579990
LOC107057684	6.18	5.55 × 10^−5^	G-type lectin S-receptor-like serine/threonine-protein kinase At4g27290
LOC102590804	6.16	2.38 × 10^−6^	nodulation receptor kinase-like
LOC102590432	6.13	1.54 × 10^−6^	probable L-type lectin-domain containing receptor kinase S.5
LOC102596031	6.08	1.24 × 10^−21^	uncharacterized LOC102601567. transcript variant X2
LOC102601567	6.08	1.15 × 10^−14^	uncharacterized protein At1g28695-like
** *downregulated DEGs* **			
LOC107058371	−10.00	4.61 × 10^−15^	chitin-binding lectin 1-like
LOC102605914	−8.96	7.09 × 10^−12^	putative UPF0481 protein At3g02645
LOC102579908	−8.39	4.85 × 10^−10^	uncharacterized protein LOC102579908
LOC102603653	−8.21	8.76 × 10^−10^	WD repeat-containing protein 48-like. transcript variant X2
LOC102594853	−8.20	3.90 × 10^−43^	Werner Syndrome-like exonuclease
LOC102596193	−8.13	5.76 × 10^−10^	uncharacterized protein LOC102596193
LOC102606330	−8.06	7.35 × 10^−25^	-
LOC102588561	−7.83	1.01 × 10^−8^	-
LOC102599813	−7.51	7.20 × 10^−8^	2-isopropylmalate synthase A-like
LOC102578236	−7.50	3.94 × 10^−8^	-
A’ vs. F’			
** *upregulated DEGs* **			
LOC102605504	7.07	6.44 × 10^−8^	probable trans−2-enoyl-CoA reductase. mitochondrial
LOC107061996	6.72	2.67 × 10^−7^	zeatin O-glucosyltransferase-like
LOC102585933	6.40	1.27 × 10^−20^	**flavonol synthase/flavanone 3-hydroxylase-like**
LOC102584318	6.38	1.69 × 10^−6^	uncharacterized LOC102584318. transcript variant X3
LOC102597028	6.31	4.30 × 10^−6^	uncharacterized protein LOC102597028
LOC107057684	6.18	8.19 × 10^−6^	G-type lectin S-receptor-like serine/threonine-protein kinase At4g27290
LOC102582531	6.08	5.30 × 10^−21^	replication protein A 70 kDa DNA-binding subunit B-like. transcript variant X11
LOC102579990	6.06	1.02 × 10^−5^	uncharacterized protein LOC102579990
LOC102591096	5.97	4.58 × 10^−6^	probable S-adenosylmethionine-dependent methyltransferase At5g38100
LOC102584503	5.90	3.28 × 10^−14^	small heat shock protein. chloroplastic
** *downregulated DEGs* **			
LOC102602896	−9.85	1.48 × 10^−14^	cytochrome P450 72A15-like
LOC102606330	−9.83	9.54 × 10^−15^	-
LOC102578543	−8.42	3.99 × 10^−14^	salicylate carboxymethyltransferase-like
LOC102603573	−7.84	1.12 × 10^−8^	uncharacterized LOC102603573. transcript variant X5
LOC102588561	−7.70	1.87 × 10^−8^	-
LOC107060242	−7.15	5.69 × 10^−7^	uncharacterized LOC107060242. transcript variant X2
LOC102584808	−7.05	9.08 × 10^−10^	intracellular ribonuclease LX-like
LOC102577909	−6.38	1.29 × 10^−14^	cysteine protease inhibitor 8-like
LOC107058413	−6.24	7.30 × 10^−5^	uncharacterized LOC107058413. transcript variant X1
LOC102595577	−6.22	1.80 × 10^−5^	cation/H(+) antiporter 18-like

^1^ A’ is considered as a control sample in this comparison, ^2^ FDR adjusted *p*-value, Bolded–gene of special interest after comparison of bulk samples A’ vs. B’ and A’ vs. F’.

**Table 4 ijms-24-01657-t004:** Glycoalkaloids recognized in the bulk samples A’, B’, and F’.

	Glycoalkaloids Frequency [%]					
Bulk	Leptinine I	Leptine II	Solasonine	Solamargine	α-Solanine	α-Chaconine
A’	2.8	5.0	4.2	8.0	27.2	52.8
B’	8.3	-	-	-	34.0	57.7
F’	6.6	-	-	-	35.5	57.9

## Data Availability

All raw and processed RNA-seq data have been deposited in the Gene Expression Omnibus [GEO] repository under the link (https://www.ncbi.nlm.nih.gov/geo/query/acc.cgi?acc=GSE186482, accessed on 27 October 2021).

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
