# Peer review of "Glycoalkaloid Composition and Flavonoid Content as Driving Forces of Phytotoxicity in Diploid Potato"

_ijms, 2023, doi:10.3390/ijms24021657_

Round 1

Reviewer 1 Report

There are some recommendations for the improving of Methodical part.

 4.1. Plant material

How leaves were collected from each of 113 individual plants. Does it mean from the bottom, middle, or top of plants?  Maybe there was something methodical? Because it correlates very for the following results of experiments. 

 4.2. Evaluation of the phytotoxicity of potato leaf extracts

How many experiment repetitions consists for the evaluation of potato leaf extracts?

 4.3. Construction of bulk samples.

 How many grams of leaves of three individual plants from groups A, B and F were selected for each bulk sample?

Reviewer 2 Report

The authors describe the the metabolic shift and transcriptional alteration mainly in FLS among potato progeny, and their contribution in different phytotoxicity responses in mustard. This discovery is interesting and important, however, I recommend some changes and experimental additions to provide a more-focused story for readers.

Issues and recommendations:

1. The different composition of glycoalkaloids were detected in bulk samples A, B and F based on Table 4: (1) higher percentage of leptinine I, alpha-solanine and alpha-chaconine, or could higher ratio of leptinine I and alpha-solanine in both B and F; (2) absence of Leptine II, solasonine and solamargine in both B and F. However, the authors did not describe sufficiently about these results, and did not discuss enough about these difference: whether difference (1) or (2) is the probable reason contributing to the phytotoxicity. It would be greatly helpful that either using mimic content and composition of glycoalkaloid for phytotoxicity evaluation against mustard, or analyzing glycoalkaloid composition of Group C which has low glycoalkaloid content but high phytotoxicity.

2. The transcription analysis showed many DEGs, indicating the metabolic alterations in both A vs B and A vs F. However, the authors did not give any data on glycoalkaloid metabolism: How many DEGs involved in the biosynthesis of glycoalkaloids? Which could be the critical gene(s) responding to the change of glycoalkaloid composition? Especially why there components were absent?

3. In most of case, one of the glycoalkaloids is leptine II, but it is leptinine II in line 192. They are two different chemicals. Please verify.

4. FLS was significantly up-regulated in both A vs B and A vs F, based on both RNA seq and quantitative analysis of bulk and individuals. But no data were presented to proved that flavonoids accumulated in Group B and F. Therefore it is hard to conclude that the low phytotoxicity in leaf extract from Group B and F. According to SoÅ‚tys-Kalina et al. 2019, I suggest that the authors could add flavonoid to leaf extract of bulk sample A to see whether the phytotoxicity is diminished. 

Reviewer 3 Report

The authors present a very interesting manuscript, but there are some important points should be revised.

Line 15. Space need after point and the start of new sentence.

Line 55-56. It is not true that “potato plants have a 55 well-developed shoot system”. Potatoes have a weak root system and, therefore, are particularly demanding to soil conditions (Striuk, 2007). The root mass of potatoes in relation to mass of the whole plant is less than that of many agricultural crops: the volume of soil encompassed by the root system of potatoes is almost 1.4 times less than that encompassed by the root system of barley, and 2.2 times less than that of sugar beet (Alsmik et al., 1979 – In Russia).

Line 120. It is not clear how many F1 individuals belong to each of groups. The number of F1 individuals should be given before the frequency of them.

Line 172. What are FLS expression and total flavonoid content in the parental clones? This information is obligatory in order to compare with F1 individuals’ characteristic and discussion the result.

Line 199. Even though the title of table 4 “Glycoalkaloids recognized in the parental clones and bulk samples A’, B’ and F’.” Glycoalkaloid composition in the parental clones are not presented. This information is obligatory in the study of segregating population and discussion the result.

Line 250. “…among progeny generated by crossing; however (Peng et al. 2019).” Perhaps some correction should be done.

Line 373. References Gniazdowska et al. 2007 and Oracz et al. 2007 are two papers about sunflower phytotoxic effect on mustard seeds not about mustard used as a cover crop in potato cultivation.

Line 393-394. What is the reason to prepare each bulk sample from the leaves of three individual plant from group A, B and F? According to Table 2, the number of F1 individuals in A, B and F groups are different and more than three ones. I think a brief explanation should be done to help the reader.

Line 544. I did not find the reference number 33 in the text

Round 2

Reviewer 2 Report

The authors' response make the manuscript better for publishing.

The title in v2: Composition is missing

The title: GAs is metabolite and FLS is gene. I don't think it is very good to say the FLS gene is involved in phytotoxicity. FLS gene is involved in FL biosynthesis and content, and FL may reduce the phytotoxicity.

The analysis of RNAseq data showed more data, but little support to make the conclusion. The RNAseq data should be helpful to answer some particular questions, like why the glycoalkaloid composition is so different between A' VS B', and A' vs F', and why the phytotoxicity changes. 

The GAs composotion of Group B and Group F are similar (Table 4), and the FL contents in both group are similar too (Figure 4). But why the effects of phytotoxicity are different (Table 2)?

The revised version is also not good support the results of the munuscript.
